# An Illustration of FY-3E GNOS-R for Global Soil Moisture Monitoring

**DOI:** 10.3390/s23135825

**Published:** 2023-06-22

**Authors:** Guanglin Yang, Xiaoyong Du, Lingyong Huang, Xuerui Wu, Ling Sun, Chengli Qi, Xiaoxin Zhang, Jinsong Wang, Shaohui Song

**Affiliations:** 1Key Laboratory of Space Weather, National Satellite Meteorological Center (National Center for Space Weather), China Meteorological Administration, Beijing 100081, China; yglyang@cma.gov.cn (G.Y.); xxzhang@cma.gov.cn (X.Z.); wangjs@cma.gov.cn (J.W.); 2Innovation Center for FengYun Meteorological Satellite (FYSIC), Beijing 100081, China; sunling@cma.gov.cn (L.S.); qicl@cma.gov.cn (C.Q.); 3Beijing Institute of Applied Meteorology, Beijing 100029, China; duxy@pku.edu.cn; 4National Space Science Center, Chinese Academy of Sciences, Beijing 100190, China; 5Beijing Key Laboratory of Space Environment Exploration, Chinese Academy of Sciences, Beijing 100190, China; 6Joint Laboratory on Occultations for Atmosphere and Climate, National Space Science Center, Chinese Academy of Sciences, Beijing 100190, China; 7Key Laboratory of Science and Technology on Space Environment Situational Awareness, Chinese Academy of Sciences, Beijing 100190, China; 8State Key Laboratory of Geo-Information Engineering, Xi’an 710054, China; 9Shanghai Astronomical Observatory, Chinese Academy of Sciences, Shanghai 200030, China; xrwu@shao.ac.cn; 10School of Resources, Environment and Architectural Engineering, Chifeng University, Chifeng 024000, China; 11Chinese Academy of Sciences, Beijing 100049, China; 12Key Laboratory of Radiometric Calibration and Validation for Environmental Satellites, National Satellite Meteorological Center (National Center for Space Weather), China Meteorological Administration, Beijing 100081, China; 13School of Remote Sensing and Geomatics Engineering, Nanjing University of Information Science and Technology, Nanjing 210044, China; 20201248063@nusit.edu.cn

**Keywords:** GNOS-R, GNSS-reflectometry, SMAP, soil moisture, vegetation, surface roughness

## Abstract

An effective soil moisture retrieval method for FY-3E (Fengyun-3E) GNOS-R (GNSS occultation sounder II-reflectometry) is developed in this paper. Here, the LAGRS model, which is totally oriented for GNOS-R, is employed to estimate vegetation and surface roughness effects on surface reflectivity. Since the LAGRS (land surface GNSS reflection simulator) model is a space-borne GNSS-R (GNSS reflectometry) simulator based on the microwave radiative transfer equation model, the method presented in this paper takes more consideration on the physical scattering properties for retrieval. Ancillary information from SMAP (soil moisture active passive) such as the vegetation water content and the roughness coefficient are investigated for the final algorithm’s development. At first, the SR (surface reflectivity) data calculated from GNOS-R is calculated and then calibrated, and then the vegetation roughness factor is achieved and used to eliminate the effects on both factors. After receiving the Fresnel reflectivity, the corresponding soil moisture estimated from this method is retrieved. The results demonstrate good consistency between soil moisture derived from GNOS-R data and SMAP soil moisture, with a correlation coefficient of 0.9599 and a root mean square error of 0.0483 cm^3^/cm^3^. This method succeeds in providing soil moisture on a global scale and is based on the previously developed physical LAGRS model. In this way, the great potential of GNOS-R for soil moisture estimation is presented.

## 1. Introduction

Over the past three decades, signal of opportunity reflectometry (SoOP-R) has attracted increasing attention, while global navigation satellite system (GNSS) reflectometry (GNSS-R) has employed the GNSS signal as the transmitted signal source. Spaceborne GNSS-R employs a specially designed GNSS-R receiver to collect signals reflected from the Earth’s surface. The working mode of GNSS-R is a multi-static radar that operates in the L-band. In comparison with the traditional microwave remote sensing technique, the main advantage of GNSS-R is that it can achieve a high spatiotemporal resolution using a small, lightweight instrument that has low power consumption [1,2].

Several space-borne GNSS-R missions have been launched over the last two decades. UK DMC was the first mission, and it demonstrated the possibility of GNSS-R for soil moisture detection [3]. The TechDemoSat-1 (TDS-1) satellite was launched successfully on 8 July 2014. The satellite is equipped with a GNSS reflected signal receiver (i.e., the SGR-ReSI; space GNSS receiver remote sensing instrument). TDS-1 provided a large number of land surface GNSS reflection data [4]. NASA’s Cyclone Global Navigation Satellite System (CYGNSS) was launched in 2016. In addition to applications in the marine field, CYGNSS data have also been used in studies on soil moisture, vegetation biomass, surface freezing and thawing, flood inundation, and wetland detection [5,6,7,8,9,10]. After the SMAP radar malfunctioned, it was tuned to collect the reflected signals of GPS (global position system) which is called SMAP-R (SMAP-reflectometry). Additionally, its applications on land surface parameter monitoring have been demonstrated [11]. The BuFeng-1 A/B twin satellites were part of the first Chinese GNSS-R satellite mission, launched in June 2019 [12]. FSSCat was launched in September 2020, and its application includes not only sea ice and sea salinity but also soil moisture [13]. There are also some other future GNSS-R missions planned such as ESA Pretty mission 3Cat-4 and HydroGNSS [14].

The FengYun-3E (FY-3E) is the fifth satellite in China’s polar-orbiting meteorological satellite series. It is the first early morning orbit satellite (descending at around 6:00 local time and ascending at around 18:00 local time) in the FengYun satellite family, and also the first such platform among the various operational meteorological satellites [15]. The FY-3E has 11 payloads. One of them is the GNSS Occultation Sounder II (GNOS-II), a new generation of GNSS-R sensor for the FY-3E meteorological satellite. Figure 1 summarizes the development of space-borne GNSS-R missions or payloads.

The National Space Science Center of the Chinese Academy of Sciences is responsible for the development of the GNOS-R payload [16]. According to the launch schedule, subsequent FY-3 satellites (i.e., 3F/3G/3H) will also be equipped with GNOS-II sensors. Soil moisture retrieval is an important scientific goal of FY-3E GNOS-R, the distinctive features of which can be summarized as follows:

The inclination of the orbit is 98.8°. GNOS-R can detect the complex and changeable land surface globally within the latitude range of ±85° with a mean revisit time of about 5.5 days. Therefore, it can provide data of nearly global coverage;GNOS-R operates at an altitude of 200–350 km higher than that of other similar payloads; therefore, it has the highest orbit (up to 836 km) of any GNSS-R mission. The differences in the vegetation, surface roughness, and other factors within the scattering area have a greater impact on the GNOS-R soil moisture inversion;GNOS-R can receive GPS (global position system) and BDS (Beidou navigation system) navigation satellite signals simultaneously. It should be noted that the reflection signals from Galileo are also available. That is to say, the GNSS signals received by the instrument are GPS L1 C/A, BDS B1I, and GAL E1B. Therefore, it can provide users with multi-GNSS reflection products.

The main objective of FY-3E GNOS-R is to study ocean parameters, while soil moisture retrieval with this new payload may be one of its significant scientific objectives. Although there are several global soil moisture datasets (Table 1), the high spatial and temporal resolutions of GNSS-R are unique. For example, the commonly used SMAP data are 36 km and with 2–3 days for a global revisit.

GNOS-R is the first GNSS-R payload on FY-3E. It will provide ocean wind products, and this is the first objective of the payload. However, soil moisture retrieval will be the second scientific objective. Therefore, this paper has tried to employ GNOS-R data for soil moisture retrieval in order to validate its possibility for this objective. There will be other GNSS-R payloads on Fengyun satellites, and if the soil moisture retrieval has been validated, the valuable products retrieved from other networks of Fengyun satellites’ GNSS-R payloads will provide a new kind of soil moisture products compared to the traditional remote sensing datasets. That is to say, with the help of GNOS-R, higher spatial and temporal resolution products of global soil moisture will be possible. Of course, this will be achieved after the networks of GNOS-R are provided. However, the first step is to perform research illustrating its possibility and methodology to demonstrate its potential for global soil moisture retrieval.

**Table 1 sensors-23-05825-t001:** Global soil moisture datasets [17].

Property	Product (Version)	Grid Resolution	Represent Depth	Reference(s)
Low-frequency passive microwave product (L-band)	SMAP..MCCA	36 km	0–5 cm	[18]
	SMAP-L2-R17000 (SCA-V\SCA-H\MDCA)	36 km	0–5 cm	[19]
				[20]
	SMOS-L2-V650	15 km		[21]
	SMOS-L3-V300	25 km	0–5 cm	[22]
	SMOS-IC-Version 2.0	25 km		[23]

As for soil moisture retrieval, several kinds of methods have been proposed. Using the previously developed land surface GNSS reflection simulator (LAGRS) model [24,25], this paper is the first to present soil moisture retrieval using FY-3E GNOS-R. Section 2 provides an introduction to the dataset. The retrieval methodology is described in Section 3. The results and the accuracy estimations are presented in Section 4. Finally, the conclusions are provided in Section 5.

## 2. FY-3E GNOS-R Dataset

The observational area of FY-3E GNOS-R covers all areas within the latitude range of ±85°. Daily observations, excluding the sea surface, are distributed on land and ice surfaces with percentages of 24% and 14%, respectively. The data are expected to be available to the public at the end of May 2022.

The specular reflection points of FY-3E GNOS-R on a single day (10 August 2021) are shown in Figure 2. The statistical results of GNOS-R original observational data at different local times are presented in Figure 3, where the simulated data are derived from a network of early morning satellites (EM), morning satellites (AM), and afternoon satellites (PM).

To utilize the final geophysical parameters’ retrieval, delay-Doppler mapping (DDM) is the original observation of GNSS reflectometry (GNSS-R). Compared to the conventional uniform DDM from the TDS-1 and CYGNSS Level 1 products, the DDM generated by the GNOS-R receiver is non-uniform to obtain more sampling near the specular point. Figure 4 provides an example of the onboard DDM [26].

In order to achieve the final soil moisture retrieval, we will employ the peak values of the DDM for calculations. The scattering geometry can be summarized as in Figure 5:

It should be mentioned that most of the present works related to soil moisture retrieval using space-borne GNSS-R, such as CYGNSS, have employed coherent scattering for the final retrieval; therefore, we also adopted this assumption when using FY-3E GNOS-R for soil moisture inversion [27,28]. To obtain surface reflectivity (SR), we use the following Equation (1):(1)$$\Gamma \left(\theta \right)=\frac{{\left(R_{r}+R_{t}\right)}^{2}\left(P_{DDM}-N\right)}{FR_{t}^{2}R_{r}^{2}4\pi }$$
where Γ is the SR, $R_{r}$ and $R_{t}$ are the distance from the specular points to the receiver and the transmitter, respectively, $P_{DDM}$ is the peak DDM power, N refers to noise, and F is the DDM BRCS-Factor (bistatic radar cross-section factor),
(2)$$F=\frac{\lambda^{2}P_{t}G_{t}G_{r}}{{\left(4\pi \right)}^{3}R_{t}^{2}R_{r}^{2}}$$
where $P_{t}G_{t}$ is the GNSS effective isotropic radiated power (EIRP) and $G_{r}$ is the receiver antenna pattern.

Figure 6 shows the calculated surface reflectivity (SR) of five days for 11–15 July 2021. The distribution of SR indicates the general trend of soil moisture and vegetation cover information over the globe. For example, data near the Amazon River in South America have high SR due to coherent scattering. In this study, as a preliminary demonstration, we used data of 20 days from 10–30 July 2021 to obtain the final retrieval.

## 3. Retrieval Methodology

Different from the commonly used artificial intelligence technique, we employed the LAGRS physical model for the direct soil moisture retrievals [17], and we used SMAP ancillary global soil moisture information as a reference [29].

With the orbit of FY-3E and the 1 Hz DDM sampling frequency, the spacing of specular points between one another on the same track is about 6 km. Under coherent scattering, the original spatial resolution of GNSO-R soil moisture at the specular point is around 1–2 km depending on the incidence angle (decided by the first Fresnel Zone) [30]. Based on a previous study [31], around 90% of data over land are coherent scattering, so the impact of data over incoherent scattering on the spatial resolution was ignored in this study. As the maximum incidence angle of GNOS-II data is 55°, no quality control on the incidence angle was applied.

Since the SMAP data show a 36 km EASE-grid projection (https://smap.jpl.nasa.gov/data/ (accessed on 1 January 2020)), we have gridded our GNOS-R data to this projection for the final retrieval, where around six specular points are used for each SMAP grid. Within each SMAP pixel, we averaged the SR and incidence angle of GNOS-R data. This kind of process has been used in several papers [32,33,34,35,36]. Although the number of specular points in each SMAP grid is not enough to provide a smooth result, this paper aims to present this methodology to demonstrate soil moisture retrieval, which could be applied by future studies when more data are used.

### 3.1. LAGRS Model

Our retrieval method is based on the LAGRS model, which is a spaceborne GNSS-R simulator designed specifically for FY-3E GNOS-R [24,25]. The LAGRS model has several modes for different geophysical land parameters, i.e., bare soil (frozen soil and thawed soil LAGRS-Soil), vegetation (LAGRS-Veg), and snow (LAGRS-Snow). The LAGRS model is based on microwave scattering models, but for the soil component, specular scattering, and diffuse scattering models are included. Specifically, the model developed by Fung and Eom is used to obtain coherent scattering [37]. To obtain the circular reflection coefficients σ, we use the wave synthesis technique while the specular scattering matrix is employed to obtain the final results.
(3)$$R\left(\mu \right)=\left[\begin{array}{cccc} {\left|\gamma_{v}\right|}^{2} & 0 & 0 & 0\\ 0 & \left|r_{h}^{2}\right| & 0 & 0\\ 0 & 0 & Re\left(\gamma_{v}\gamma_{h}^{*}\right) & -Im\left(\gamma_{v}\gamma_{h}^{*}\right)\\ 0 & 0 & Im\left(\gamma_{v}\gamma_{h}^{*}\right) & Re\left(\gamma_{v}\gamma_{h}^{*}\right) \end{array}\right]$$
where the subscript v and h are the vertical and horizontal polarization, and γ is the Fresnel reflectivity. Re and Im are used to obtain the real part and the imaginary part.

Other random surface scattering models such as the physical optics (PO) model, the geometrical optics (GO) model, the small perturbation method (SPM), and the integral equation model (IEM) or advanced integral equation model (AIEM) are employed to obtain the diffuse scattering properties [38]. The corresponding LAGRS models are marked as LAGRS-GO, LAGRS-PO, LAGRS-SPM, and LAGRS-IEM (LAGRS-AIEM). In order to clarify the application limitation of the different models, Table 2 summarizes the valid and recommended conditions [38].

The dielectric constant model is used to connect soil moisture and the soil dielectric constant [39].

The reflected GNSS signal power is composed of coherent scattering and incoherent scattering [40]. The formula to calculate the reflected signal power can be expressed as:(4)$${\langle \left|P_{pq}\left(\tau ,f\right)\right|\rangle }^{2}={\left|P_{pq}^{coh}\left(\tau ,f\right)\right|}^{2}+{\left|P_{pq}^{ncoh}\left(\tau ,f\right)\right|}^{2}$$
where ${\left|P_{pq}\left(\tau ,f\right)\right|}^{2}$ is the coherent scattering term; ${\left|P_{pq}^{coh}\left(\tau ,f\right)\right|}^{2}$ is the incoherent scattering term; and τ and f represent the time delay and the Doppler shift, respectively. With the LAGRS-Soil model, we can obtain the theoretical DDM of any combination of soil moisture and observation geometry, and it assists in the final soil moisture retrieval.

### 3.2. LAGRS-Based Soil Moisture Retrieval

The flowchart of the soil moisture retrieval process is presented in Figure 7. First, we need to calibrate the FY-3E GNOS-R data. In terms of the incident angle, we limit our data to angles lower than 65°, which is a range widely adopted in CYGNSS angle limitation. In terms of the signal-to-noise ratio (SNR), we pass over all the SNRs lower than −5 dB [41]. This part is often called quality control (QC). At present, we give a very simple method for this process, and other more complex methods of QC may improve the final soil moisture retrieval accuracy, such as altitude control, as for CYGNSS soil moisture application, elevations higher than 600 m are not used. Here, we employ the ratio between theoretical values (obtained from LAGRS) and measured values to derive the calibration factor (CF) [29], the process of which is described in detail in Section 4.1. Other kinds of calibration methods for GNOS-R are welcome and ongoing. Following calibration, we can use the data for retrieval. However, we need to consider the effects of soil roughness and vegetation because both can affect the accuracy of the final retrieval. Using SMAP ancillary data, we can obtain information on soil moisture and soil texture (clay content). By employing the coherent scattering component of LAGRS, we can realize roughness and vegetation information, denoted as the RV-Factor in the retrieval process. It should be noted this cannot totally correct the surface roughness and vegetation effects, but we want to reduce the effects of these two factors. With the help of SMAP data, we can estimate the effect of surface roughness and vegetation. This method is highly dependent on SMAP data. Of course, other ancillary data of the soil moisture and texture can also be applicable to this method. Using the RV-Factor, we can obtain the global surface Fresnel reflectivity, i.e., the mask of Fresnel reflectivity. In this way, soil moisture information can be related directly to the mask of Fresnel reflectivity, and we can retrieve the final soil moisture estimation: since the Fresnel reflectivity is related to the soil dielectric constant, which can be calculated by our model of LAGRS, we set up a look-up table and then the soil moisture information was retrieved.

## 4. Results and Analysis

Using the method described in Section 3, we calculated the corresponding factors, and we used these factors to realize the final retrieval results that are presented in this section.

### 4.1. FY-3E GNOS-R Calibration Factor

Calibration is the basic premise and guarantee of quantitative soil moisture inversion. During the actual operation, taking the ground target as the reference benchmark and performing indirect detection based on outfield work is an established and effective method of calibration [40]. For the calibration in GNOS-R, we selected natural feature targets with certain characteristics to calibrate the SR. The surface roughness coefficients and the vegetation water content derived from SMAP were selected as ancillary data and treated as the measured SR, while we employed the LAGRS model to obtain the theoretical SR. The theoretical values of the calibrated targets (such as water and desert) are compared with the measured values. In this way, we can derive the CF. The flowchart of the process for calculating the CF is shown in Figure 8.

The test areas used for calibration are either water bodies or bare soil. For the former, we used Lake Tahoe and Lake Qinghai, and we assumed that the reflectivity in these areas is close to 1. This is because the two areas are covered by water, whose surface reflectivity can be treated as total specular scattering. For the latter, we used desert areas and assumed that roughness effects during the calibration process could be ignored since the data time coverage for our analysis is about one month. The roughness coefficient in these ranges is almost constant. Details regarding the test areas are presented in Table 3, where latmin and latmax represent the minimum and maximum latitudes, respectively, and longmin and longmax represent the minimum and maximum longitudes, respectively.

Figure 9 presents the land cover and land use of the test areas. To illustrate the data clearly, we set the area of coverage to be larger than that of the final tested data. We can see the land cover type for the Bolivia site (ID1) is mostly low vegetation; therefore, the environment of the test field should be as homogeneous as possible. We also selected part of the Sahara Desert (ID2) as a test site, for which the land cover is bare soil, and its roughness effects can be ignored. The third and fourth test areas comprise low vegetation surrounding lakes. The water area of both Lake Tahoe (ID3) and Lake Qinghai (ID4) can be clearly identified in Figure 9. The central position of Lake Tahoe is 39°05′ N, 120°02′ W, while that of Lake Qinghai is 36°53′ N, 100°16′ E.

Using the process illustrated in Figure 7 and the information provided in Table 3, we employed the SR data of FY-3E GNOS-R for calibration. Figure 10 presents the histograms of SR data before and after calibration for the four types of geophysical land cover.

### 4.2. Roughness–Vegetation Factor

Surface roughness and vegetation cover affect both the surface bistatic scattering and the coherent scattering energy. With the function of the vegetation module of LAGRS, i.e., LAGRS-Veg, we simulated and analyzed the effects of vegetation parameters on the final scattering of LR polarization [42]. Figure 11 presents the simulation results of different components on the final scattering power [43]. While the input of the simulation is a tall Aspen stand, the total scattering of this forest canopy is dominated by the trunk component, which will have strong scattering in the specular direction. The second term which dominates the total scattering is the specular ground term. The other two components of the scattering are from the random rough surface scattering and forest crown parts. Although this is only one illustration of the forest canopy, it can tell us that the final scattering in the specular direction comprises not only the effects of bare soil (soil moisture) but also vegetation and surface roughness effects. Both are important factors that affect the accuracy of soil moisture inversion.

Therefore, during our retrieval, the RV-Factor is employed to account for the effects of surface roughness and vegetation attenuation. To remove these effects, we employed the FY-3E GNOS-R’s LAGRS model. There are two important elements related to bare soil retrieval. With these models, we can obtain the RV-Factor of the corresponding FY-3E GNOS-R. The RV-Factor can provide a basic idea of the surface roughness and vegetation information on a global scale. The global land cover type is shown in the top panel of Figure 12, and the global surface roughness coefficients, which are obtained from the SMAP L3 data, are shown in the bottom panel [29].

From the RV-Factor map shown in Figure 13, it can be seen that as the surface roughness increases, the RV-Factor increases. Additionally, as the vegetation canopy becomes denser, the RV-Factor increases. As the surface roughness increases, scattering in the coherent direction decreases, whereas diffuse scattering increases. When we take vegetation into consideration, denser vegetation (e.g., high forest canopy) causes the attenuation of the reflected GNSS signals in the L-band to increase, which causes the apparent RV-Factor to change, as can be seen in Figure 13.

Using the RV-Factor, we can obtain the mask of Fresnel reflectivity, which means that the roughness effects and the vegetation effects can be removed using the abovementioned method. The final mask of Fresnel reflectivity affects only the target soil moisture information. Figure 14 presents the mask of Fresnel reflectivity. The correlation coefficient between the FY-3E GNOS-R Fresnel reflectivity and the SMAP-derived Fresnel reflectivity is as high as 0.92.

### 4.3. Retrieval Results and Accuracy Estimation

After obtaining the mask of Fresnel reflectivity, we can use it to retrieve soil moisture by employing the LAGRS-DC and LARGS-Coh components. The global soil moisture information retrieved from FY-3E GNOS-R for July 2021 is shown in Figure 15. We also calculated the soil moisture for the same period using the SMAP data. We found reasonable agreement between the GNOS-R soil moisture and the SMAP soil moisture, with a correlation coefficient of >0.90.

The accuracy estimation for the FY-3E GNOS-R soil moisture retrieval is shown in Figure 16. The horizontal axis is the retrieved soil moisture, and the cumulated SMAP soil moisture is presented on the vertical axis. There is a reasonable correlation between the two datasets (correlation coefficient: 0.96, root mean square error: 0.048 cm^3^/cm^3^).

The main advantage of FY-3E GNOS-R for soil moisture retrieval is that it can cover high-latitude areas, which cannot be covered by CYGNSS. This will provide unique global soil moisture retrieval with space-borne GNSS-R missions. Aside from GNOS-R, SMOS and SMAP are two traditional L-band radiometry missions. It should be mentioned that these two missions will end their design life by 2024. The global measurements of GNOS-R and its follow-on missions of the FY satellite series will provide some continuity between existing and future L-band missions [44,45].

### 4.4. Discussion

At present, there is only one GNSS-R payload on the FY-3E satellite with about a 5-day revisit time, but with the launch of more payloads on future FY satellites, this will provide global soil moisture monitoring with high spatial and temporal resolutions. This will compensate for the present soil moisture products of SMAP. Meanwhile, the scattering mechanism of GNOS-R is different from that of SMAP; the former is forward scattering and the latter is passive radiometer and active radar, although the radar is out of work in its backscattering mode. SMAP has been tuned to receive the GNSS reflected signals, i.e., SMAP-R [43]. The bistatic scattering properties of GNOS-R and the emissivity of the SMAP radiometer can provide different electromagnetic wave information due to the obviously different observation geometry. The main difference between GNOS-R and SMAP-R is their polarization properties; the latter uses HR and VR polarization, which are very different from the present GNOS-R, CYGNSS, and other GNSS-R missions. However, the future GNOS-R payloads on FY-series satellites will have the potential of full polarization [41], which will benefit the vegetation and soil moisture monitoring. With the help of the fully physical scattering models designed oriented to GNOS-R, i.e., the LAGRS model, the scattering properties of different land cover types, such as desert, highly forested, or around water, can be analyzed in detail and the corresponding retrieval methods for these land types will be presented in our following work.

## 5. Conclusions

Most of the present works about soil moisture retrieval from space-borne GNSS-R missions are related to CYGNSS, whose coverage is around ±38°. With the launch of the FY-3E GNOS-R, soil moisture retrieval from the global scale GNSS-R payload becomes possible. This paper presents global results obtained from FY-3E GNOS-R. One unique feature of the retrieval method described is based entirely on a physical LAGRS model, which is a physical mechanism model designed for the GNOS-R sensor. Therefore, this method can use only a smaller amount of data (here, only about twenty days of GNOS-R Level 1 data) for the final demonstration. Using this method, we directly retrieved global soil moisture information using ancillary data from SMAP. With the calibration method employed for CYGNSS, the calibration and removal of the effects of surface roughness and vegetation oriented to GNOS-R are presented at first. The FY-3E GNOS-R Fresnel reflectivity demonstrated a strong correlation with the SMAP-derived Fresnel reflectivity (correlation coefficient: 0.92). The final soil moisture retrieval results were compared with the SMAP data for the same period of time. The correlation coefficient between them was found as high as 0.96, and the root-mean-square error between the two datasets was 0.048 cm^3^/cm^3^. The present results are sufficient to show the possibility of GNOS-R for global soil moisture retrieval, which will increase the potential scientific objectives of GNOS-R application except for ocean winds. However, it should be mentioned that the entire retrieval methodology is highly dependent on SMAP data. The development of a soil moisture retrieval algorithm is completely independent of other ancillary data and validation against in situ data will be the subject of our future work.

## Figures and Tables

**Figure 1 sensors-23-05825-f001:**
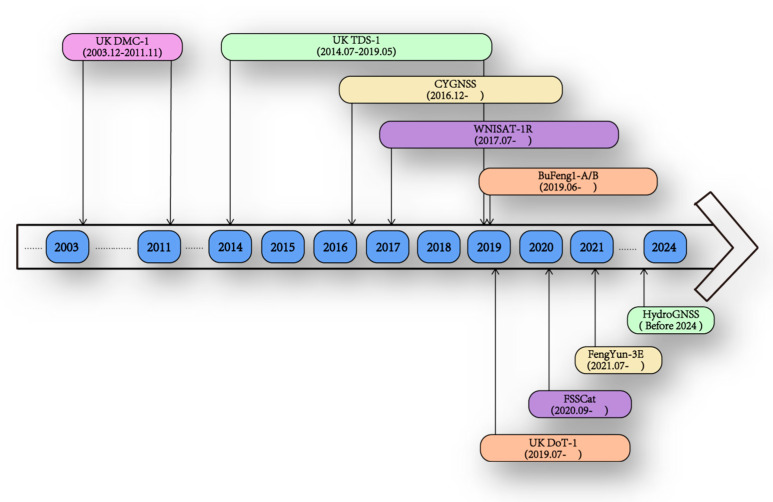
Development of space-borne GNSS-R missions.

**Figure 2 sensors-23-05825-f002:**
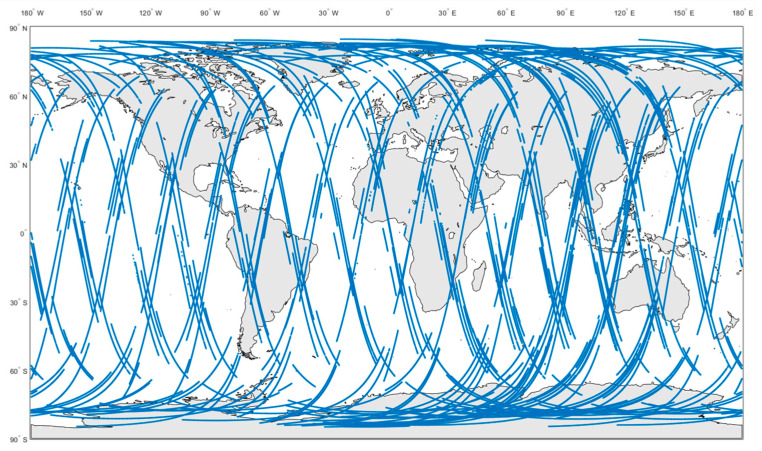
Specular reflection points of FY-3E GNOS-R on 10 August 2021.

**Figure 3 sensors-23-05825-f003:**
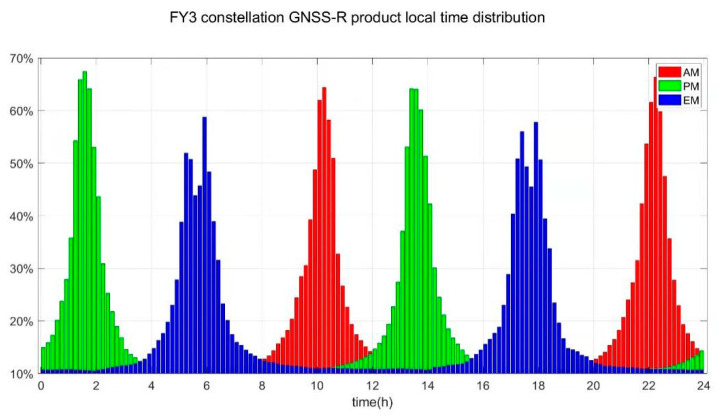
Simulated statistical data distribution of GNOS-R constellation observational data at different local times (early morning satellite (EM), morning satellite (AM), and afternoon satellite (PM)) where “early morning”, “morning” and “afternoon” stand for the descending time of the satellite.

**Figure 4 sensors-23-05825-f004:**
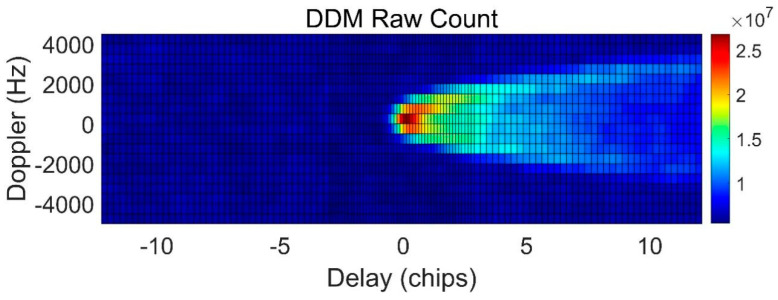
FY3E GNSS-R non-uniform DDM measurement.

**Figure 5 sensors-23-05825-f005:**
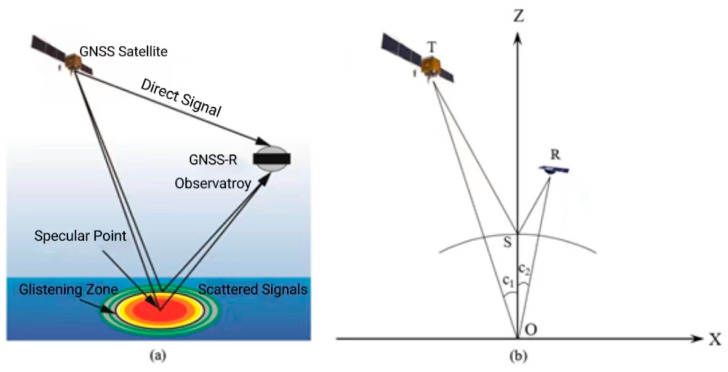
GNSS-R scattering geometry (**a**) and specular coordinate system (**b**).

**Figure 6 sensors-23-05825-f006:**
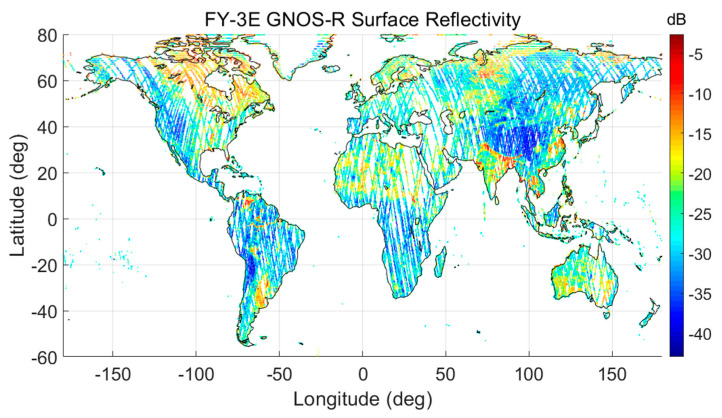
FY-3E GNOS-R surface reflectivity during 11–15 July 2021.

**Figure 7 sensors-23-05825-f007:**
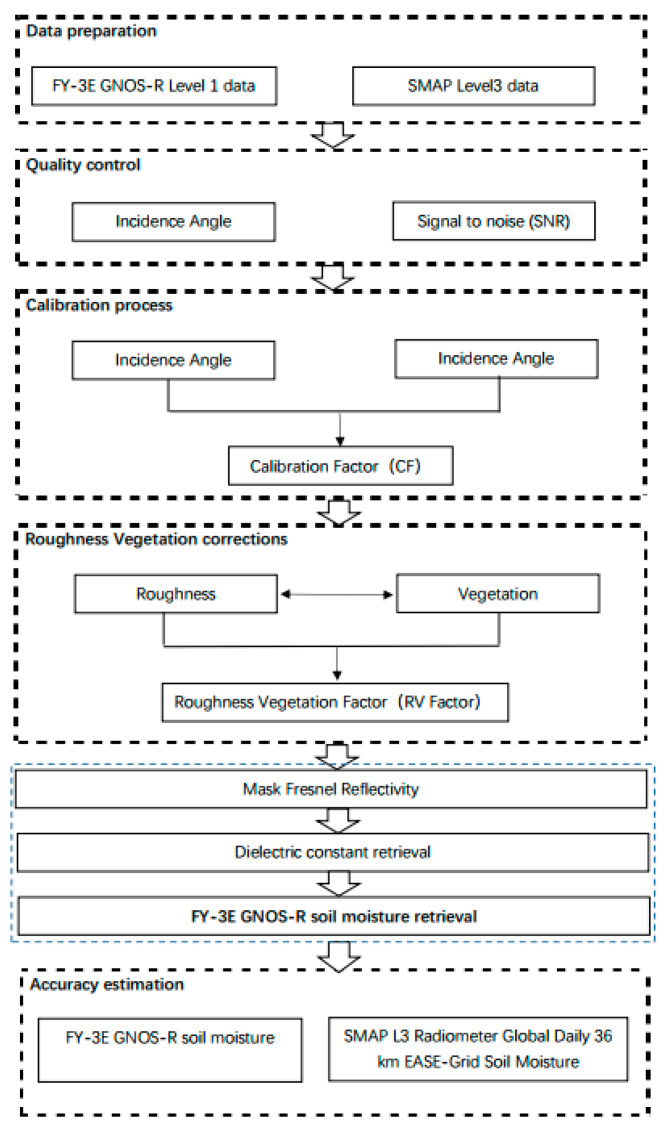
Flowchart of FY-3E GNOS-R soil moisture retrieval process.

**Figure 8 sensors-23-05825-f008:**
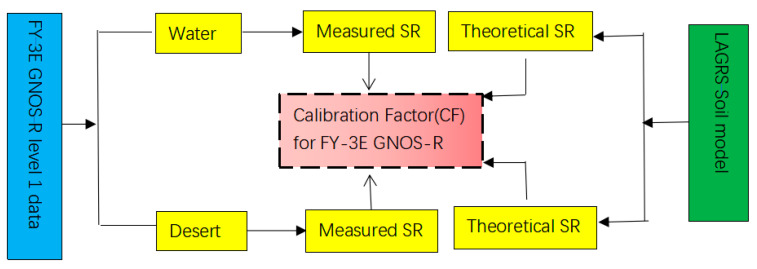
Flowchart of process for calculating the calibration factor (CF).

**Figure 9 sensors-23-05825-f009:**
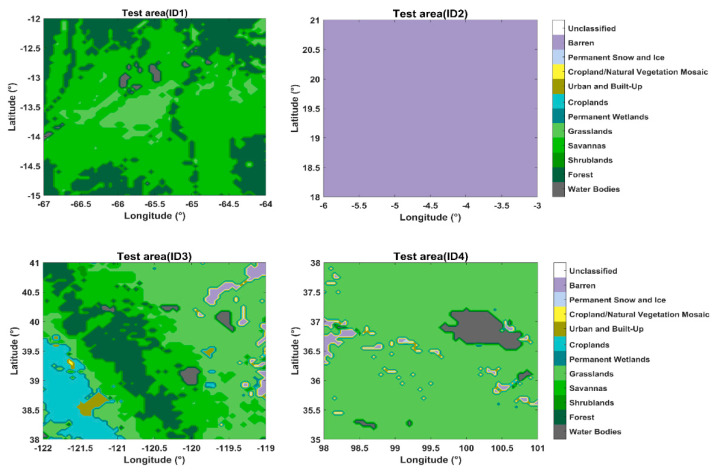
IGBP of the test areas.

**Figure 10 sensors-23-05825-f010:**
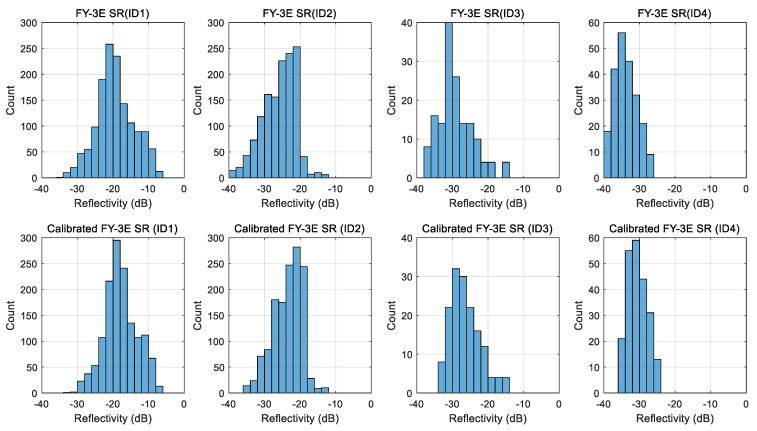
Histograms of surface reflectivity (SR) data before and after calibration for the four types of geophysical land parameters.

**Figure 11 sensors-23-05825-f011:**
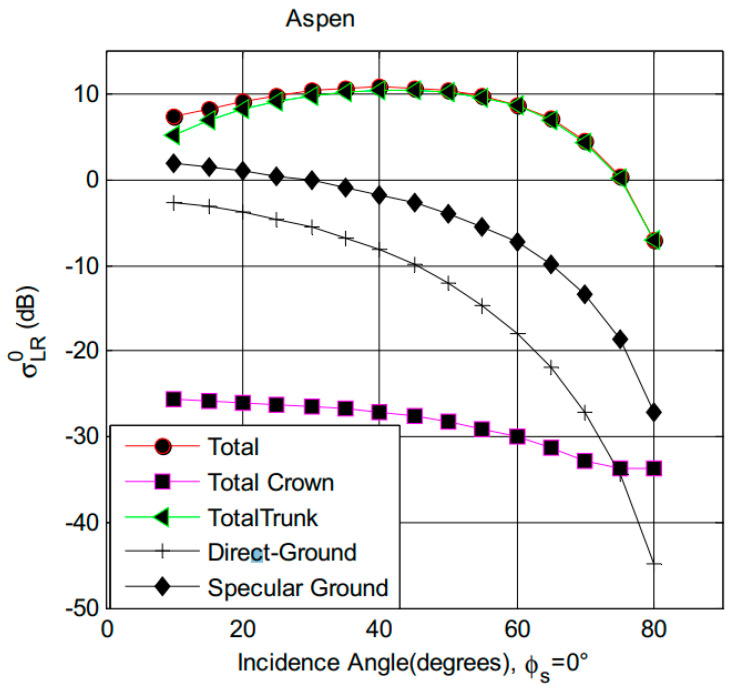
Specular scattering of different components contributions for LR polarization [34].

**Figure 12 sensors-23-05825-f012:**
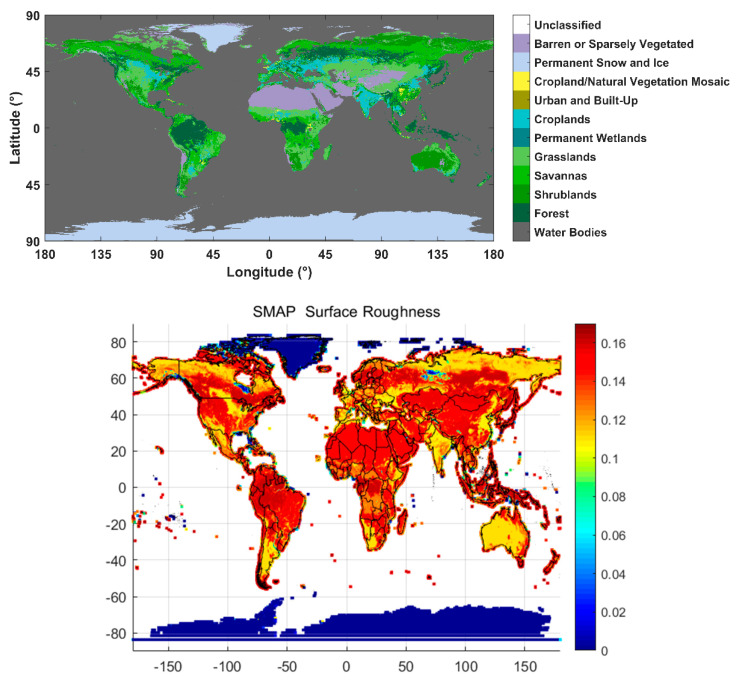
Global vegetation type and land cover (**top**) and global roughness coefficients (**bottom**).

**Figure 13 sensors-23-05825-f013:**
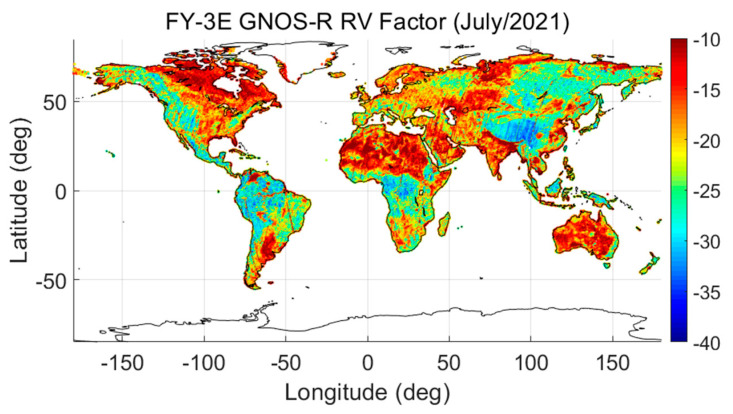
Roughness–vegetation factor (RV-Factor) derived from the LAGRS model.

**Figure 14 sensors-23-05825-f014:**
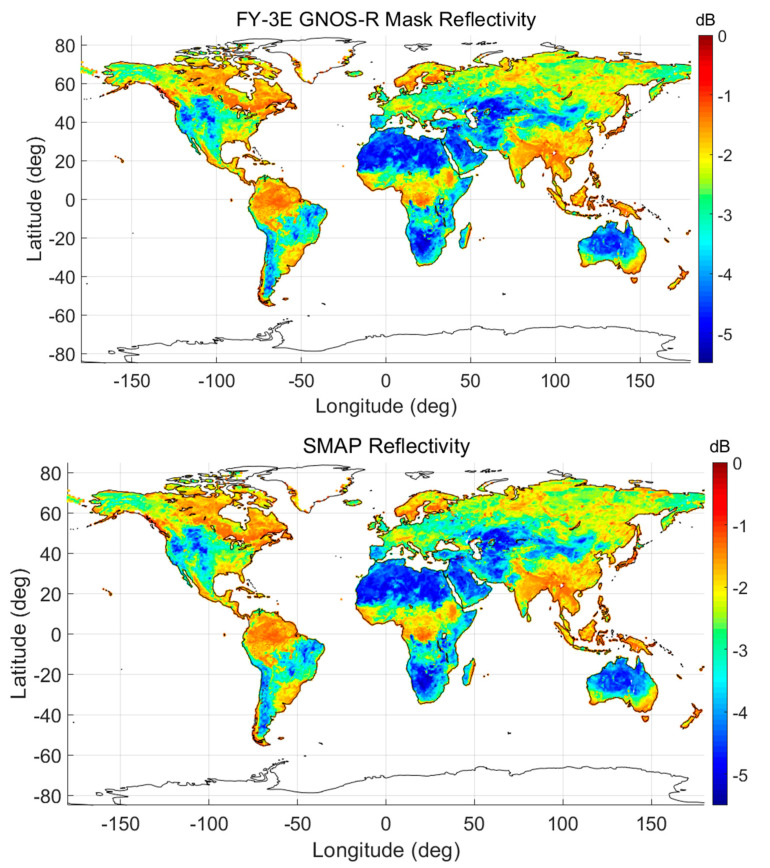
Mask of Fresnel reflectivity derived from FY-3E GNOS-R (**top**) and Fresnel reflectivity obtained from SMAP data (**bottom**) for data of 10–30 July 2021.

**Figure 15 sensors-23-05825-f015:**
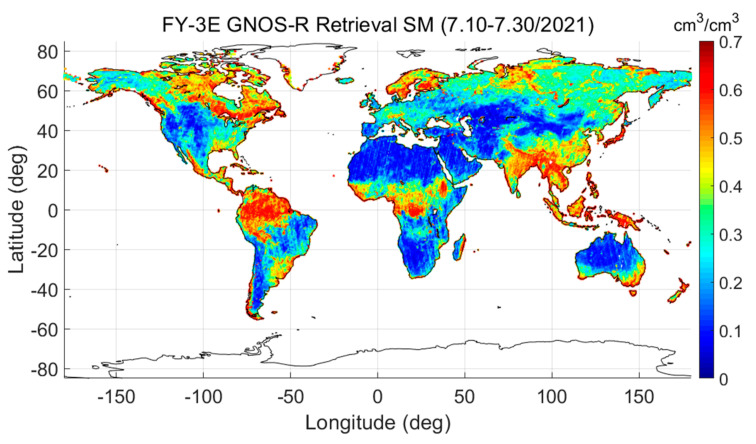
Global soil moisture retrieved from FY-3E GNOS-R from 10 July to 10 August in year 2021.

**Figure 16 sensors-23-05825-f016:**
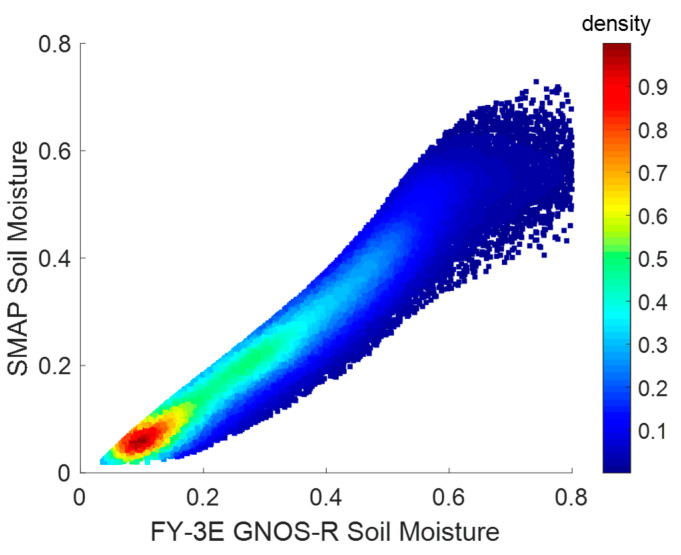
Correlation between FY-3E GNOS-R soil moisture and SMAP soil moisture for the period 10–30 July 2021 (correlation coefficient: 0.96, root mean square error: 0.048 cm^3^/cm^3^).

**Table 2 sensors-23-05825-t002:** Valid and recommended conditions for different LAGRS models, where m is the surface root-mean-square slope, $l_{s}$ is the surface correlation length, and λ is the GNSS wavelength.

Model Name	Valid Conditions			Recommended Conditions	
LAGRS-GO	s≥λ/3	$l_{s}\geq \lambda$	$l_{s}^{2}>2.76s\lambda$	0.4≤m≤0.7	
LAGRS-PO	0.05λ≤s≤0.15λ	$l_{s}\geq \lambda$	m≤0.25	$l=\lambda /6,l_{s}\geq 6l$	0.05λ≤s≤0.15λ
LAGRS-SPM	s≤0.05λ	m≤0.3	$l_{s}\leq 0.5\lambda$	$l_{s}\leq 0.25\lambda$	s≤0.05λ
LAGRS-IEM	√	√	√	√	√
LAGRS-AIEM	√	√	√	√	√

**Table 3 sensors-23-05825-t003:** Details of the location of the calibration areas.

ID	Location, Country	Latmin	Latmax	Longmin	Longmax
1	Bolivia	−15	−12	−67	−64
2	Sahara	18	21	−6	−3
3	Lake Tahoe	38	41	−122	−119
4	Lake Qinghai	35	38	98	101

## Data Availability

Data is unavailable due to privacy.

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
