# Peer review of "An Illustration of FY-3E GNOS-R for Global Soil Moisture Monitoring"

_sensors, 2023, doi:10.3390/s23135825_

Round 1

Reviewer 1 Report

The authors present an example of FY-3E GNOS-R for global soil moisture monitoring. The topic of the manuscript seems relevant for publishing in sensors.  The results showed good consistency between soil moisture derived from GNOS-R data and SMAP soil moisture. The validation of FY-3E GNOS-R for global soil moisture monitoring is limited in this manuscript. Authors should validate their results with other methods that could be beneficial for the scientific community to utilize FY-3E GNOS-R global soil moisture.

- The author should highlight the objectives of this paper in the abstract section.

-Please add more literature on global soil moisture datasets in a table form and the author should also highlight why FY-3E GNOS-R for global soil moisture was selected for the current study.

-How author calculate global vegetation type and land cover, and global roughness coefficients? There is no methodology presented in the manuscript.

-Why author select this specific area for IGBP?

- Please on adding some significant information in the discussion of results. 

Author Response

Reviewer 1 

Comments and Suggestions for Authors

The authors present an example of FY-3E GNOS-R for global soil moisture monitoring. The topic of the manuscript seems relevant for publishing in sensors.  The results showed good consistency between soil moisture derived from GNOS-R data and SMAP soil moisture. The validation of FY-3E GNOS-R for global soil moisture monitoring is limited in this manuscript. Authors should validate their results with other methods that could be beneficial for the scientific community to utilize FY-3E GNOS-R global soil moisture.

  1. The author should highlight the objectives of this paper in the abstract section.

Response: Thank you very much for your comments and valuable suggestions which are very helpful for our improvement of the manuscript. We have given the following information for the revision.

The main objective of GNOS-R is for ocean parameters study, while soil moisture retrieval with this new payload maybe one of its significant scientific objectives. We will illustrate its possibility and methodology to demonstrate its potential for global soil moisture retrieval.

  1. Please add more literature on global soil moisture datasets in a table form and the author should also highlight why FY-3E GNOS-R for global soil moisture was selected for the current study.

Response: Thank you very much, we have revised our paper according to your suggestions.

The global soil moisture datasets are presented in the following table: In this table, 25 datasets of global soil moisture are presented.

Meanwhile, we have revised this part and give the following demonstration in the revision like the following:

The main objective of FY-3E GNOS-R is for ocean parameters study, while soil moisture retrieval with this new payload maybe one of its significant scientific objectives. Although there several global soil moisture datasets (Table 1), the high spatial and temporal resolutions of GNSS-R is unique. For example, the commonly used SMAP data is 36 km and with 2-3 days for a global revisit.

GNOS-R is the first GNSS-R payload on FY-3E, it will provide ocean wind products and this is the first objective of the payload. However, soil moisture retrieval will be the second scientific objective. Therefore, this paper has tried to employ GNOS-R data for soil moisture retrieval in order to validate its possibility for this objective. There will be other GNSS-R payloads on Fengyun satellites and if the soil moisture retrieval has been vali-dated and then the valuable products retrieved from other networks of Fengyun satellites’ GNSS-R payloads will provide a new kind of soil moisture products compared to the traditional remote sensing datasets. That is to say, if with the help of GNOS-R, higher spatial and temporal resolutions products of global soil moisture will be possible. Of course, this is be achieved after the networks of GNOS-R are provided. However, the first step for this  is to do research on illustrating its possibility and methodology to demonstrate its potential for global soil moisture retrieval.

5 remote sensing retrieval products based on a single satellite sensor. i.e. L-band low-frequency passive microwave products are summarized in a table.

                     Table 1 Global soil moisture datasets 

Property

Product (Version)

Grid resolution

Represent depth

Reference(s)

Low-frequency passive microwave product(L-band)

SMAP..MCCA

36km

0-5cm

Zhao.et.al. (2021)

SMAP-L2-R17000

(SCA-V\SCA-H\MDCA)

36km

0-5cm

Entekhabi et al. (2010)

O’Neill et al. (2020)

SMOS-L2-V650

15km

Kerr et al. (2017)

SMOS-L3-V300

25km

0-5cm

Al Bitar et al. (2017)

SMOS-IC-Version 2.0

25km

Wigneron et al. (2021)

  1. How author calculate global vegetation type and land cover, and global roughness coefficients? There is no methodology presented in the manuscript.

Response: Thank you very much for your comments. We have employed the IGBP to classify the global land cover and land type, which can be seen in the top figure of Figure 11. In this way, we calculate the corresponding information. As for the global roughness coefficients, we do not calculate this parameter independently. We have calculated the Roughness–vegetation factor (RV-Factor) derived from the LAGRS model as shown in Figure 12. The RV factor contains not only vegetation information, but also the surface roughness information. With the RV-Factor, we can remove both the vegetation and surface roughness factors from the SR (Surface reflectivity) at the same time. In this way, we can get the pure Fresnel reflectivity from which we can retrieve the soil moisture information directly. Figure 6 presents the methodology of FY-3E GNOS-R soil moisture retrieval process. While the part correlated with the problems mentioned in this comment is plotted with yellow color like the following:

Figure 6. Flowchart of FY-3E GNOS-R soil moisture retrieval process.

  1. Why author select this specific area for IGBP?

Response: Thank you very much. the reasons and standards for the selection of the specific area for IGBP are like the following:

The surface roughness component and other attenuation factors on the coherent component of the reflected signal are both regarded as unimportant in the calibration technique employing reflectivity acquired from aquatic bodies. Here, our initial supposition entails taking into account various land surface types that display comparable diffusion properties, i.e., with a particular specular behavior and known dielectric properties, while avoiding attenuation factors on the coherent component of the signal, such as BSR (bare soil roughness) and VOD (vegetation optical depth). As a result, particular zones ought to be investigated as prospective surfaces for GNSS-R calibration and/or validation needs. One of the authors in our manuscript have participated in the research work of this topic using CYGNSS data, while the detail information can be found in the following reference and we have employed this same method for the FY-3E GNOS-R datasets as shown in this paper.

Molina, I.; Calabia, A.; Jin, S.; Edokossi, K.; Wu, X. Calibration and Validation of CYGNSS Reflectivity through Wetlands’ and Deserts’ Dielectric Permittivity. Remote Sens. 2022, 14, 3262. https://doi.org/10.3390/rs1414326

  1. Please on adding some significant information in the discussion of results.

Response: Thank you very much for your suggestions. and we have revised the conclusion section according to your comments like the following, while the sentences in blue color are the revised parts of the new conclusion.

Most of the present works about soil moisture retrieval from space-borne GNSS-R mission is related to CYGNSS, whose coverage is between ±38°.  With the launch of FY-3E GNOS-R, soil moisture retrieval from global scale GNSS-R payload becomes possible. This paper presents global results obtained from FY-3E GNOS-R. One unique feature of the retrieval method described is based entirely on physical LAGRS model, which is a physical mechanism model designed for the GNOS-R sensor . Therefore, this method can use only a smaller amount of data (here only about twenty days of GNOS-R Level 1 data) for the final demonstration. T Using this method, we directly retrieved global soil moisture information using ancillary data from SMAP. With the calibration method that employed for CYGNSS, the calibration and removal of the effects of surface roughness and vegetation oriented to GNOS-R is presented at first, the FY-3E GNOS-R Fresnel reflectivity demonstrated a strong correlation with the SMAP-derived Fresnel reflectivity (correlation coefficient: 0.92). The final soil moisture retrieval results were compared with the SMAP data for the same period time. The correlation coefficient between them was found as high as 0.96, and the root-mean-square error between the two datasets was 0.048 cm3/cm3.  The present results are sufficient enough to show the possibility of GNOS-R for global soil moisture retrieval, which will increase the potential scientific objectives of GNOS-R application except for ocean winds. However, it should be mentioned that the entire retrieval methodology is highly dependent on SMAP data. The development of a soil moisture retrieval algorithm completely independent of other ancillary data and validation against in-situ data will be the subject of our future work.

Author Response

Reviewer 2

This paper is showing the soil moisture retrieval results using the FY-3 GNOS instrument. The results show a very good correlation with the SMAP product. The paper is well organized. There are several points need to be clarified.

  1. Please denote the frequency this GNSS-R instrument is working at. Is it L band around 1.2GHz~1.5GHz?

Response: Thank you very much for your comments. The GNSS signals received by the instrument are GPS L1 C/A, BDS B1I and GAL E1B. We give the following description in the revision.

It should be noted that the reflection signals from Galileo is also available. That is to say, the GNSS signals received by the instrument are GPS L1 C/A, BDS B1I and GAL E1B. Therefore, it can provide users multi-GNSS reflection products.

  1. Line 120-123. The manuscript describes a denser sampling near the specular point. It would be better to provide an estimation of the physical area on the ground for the ddm bins near the specular point.

Response: Thank you very much. The coherent integration time and incoherent integration time is 1ms and 1s, respectively. The physical area corresponds to an instant 1ms DDM is around 1km1km. The physical area corresponds to the final averaged DDM is around 1km6km due to the incoherent integration.

  1. This may be out of the scope of the paper. But I’m wondering how well the ddm bins near the specular point independent from each other are?  

Response: Thank you very much for your comment. All bins in the DDM are sampled independently in the receiver.

  1. Are you using the peak value in the DDMs for retrieval? Please provide some details on that particular DDM bin.(e.g. effective area, delay and doppler)

Response: Thank you very much for your comment. We have provided more information for this part in the revision like the following:

In order for the final soil moisture retrieval, we will employ the peak values of the DDM for the following calculation.

Yes, we use the peak value in the DDM for retrieval. The peak bin corresponds to the specular point when it is over land, and thus its delay and doppler are zero. The effective area is around 1km 6km.

  1. Need a figure for the geometry of the GNSS-R scattering. Then the readers may have better understanding of the signal scattering.

Response: Thank you very much for your suggestion. A sentence for explanation is added:

In order for the final soil moisture retrieval, we will employ the peak values of the DDM for the following calculation. While the scattering geometry can be summarized like the following:

We have provided a new figure for the geometry of the GNSS-R scattering:

Figure GNSS-R scattering geometry (a) and specular coordinate system (b)

  1. It would be great to show more details on the forward model (LAGRS model) of the GNSS-R system. It is not clear to me how you treated the scattering of land surface. Also as in the paper, the LAGRS model includes the scattering of coherent and incoherent. It seems to me that you are treating the land surface majorly as coherent scattering. Based on my knowledge up to now, there’s almost no coherent scattering on the land surface except for several locations. But it is ok for you to use the coherent scattering model for retrieval. Just need to clarify what kind of scattering you are using.

Response: Thank you very much for your comment and suggestion. LAGRS model is a gradually improved land surface satellite borne DDM simulator. It is a systematic model, with some of its functions specifically developed for the FY-3E GNOS-R payload. During the calculation for soil moisture retrieval as shown in this paper, we have employed the coherent scattering for the final utilization. The corresponding references are presented in the revision. While the ultimate development goal of LAGRS model is to make it a mechanism model based on microwave scattering model, which can be used to calculate the spaceborne GNSS-R output of typical land geophysical parameters (soil layer covered by vegetation layer). The LAGRS model will have the calculation abilities of multifrequency and multisystem for Beidou, GPS, Galileo and GLONASS satellites. Moreover, the model can also calculate coherent (LAGRS-coh model) and incoherent (LAGRS-non model) scattering characteristics in different observation geometries (scattering zenith angle and azimuth angle). LAGRS will also have full polarization capability, which means that it can be used in calculations for various polarizations, e.g., circular and linear (LAGRS-pol module). All typical land surface parameters can be covered using this simulator, which include but are not limited to soil moisture (LAGRS-Soil) (Wu and Xia, 2021), vegetation (LAGRS-Veg), snow (LAGRS-Snow), soil freeze/thaw process (LAGRS-Soil F/T), wetland (LAGRS-Wetland), and flood inundation (LAGRS-Flood).

The LAGRS simulator is generally based on the ocean surface GPS scattering model, which is the integral form of the bistatic radar equation. However, as for the vegetation case, the reflected GNSS signal power is composed of coherent scattering and incoherent scattering from the glistening region. The reflected signal power can be expressed as follows:

                              (13)

where  is the power generated by specular reflection, i.e., the coherent scattering term;  is the power generated by diffuse reflection, i.e., the incoherent scattering term;  and represent the time delay and the Doppler shift, respectively.

References:

Wu, X., & Wang, F. (2023, April 14). LAGRS-Veg: a spaceborne vegetation simulator for full polarization GNSS-reflectometry. GPS Solutions, 27(3). https://doi.org/10.1007/s10291-023-01441-5

Wu, X.R., Xia, J.M., 2021. A LAND surface GNSS reflection simulator (LAGRS) for FY-3E GNSS-R payload Part I. Bare soil simulator. IEEE ASME Trans. Mechatron. 90–92.https://doi.org/10.1109/GNSSR53802.2021.9617672.

  1. I’m wondering how the DDMs are averaged. Any information or reference provided will be appreciated.

Response: Thank you very much for your comment. As for the DDM, we have employed the peak values of the DDMs (surface reflectivity) to get the soil moisture retrieval information. While the way to get the peak values of DDMs is like the following:

To obtain surface reflectivity (SR), we use the following equation[1]:

where  is the SR,  and  are the distance from the specular points to the receiver and the transmitter, respectively,  is the peak DDM power,  refers to noise, and  is the DDM BRCS-Factor (Bistatic Radar Cross Section Factor),

Where  is the GNSS Effective Isotropic Radiated Power (EIRP) and  is the receiver antenna pattern.

All the values for equation 1 and 2 to get SR can be achieved from the user guidance of FY-3E GNOS-R.

  1. This is out of the scope of this paper. I have a suggestion on the data analysis. If you can look into the raw IF data, would it be worthwhile to perform some analysis on the “coherence” of the data collected? You don’t need to include it in this paper, this is just a suggestion.

Response: Thanks for the suggestion and it is a really very valuable comment for our study. We do have raw IF data from FY-3E. We will perform coherence analysis in the future.

  1. SMAP L3 data. I’m not familiar with this level of product. What is the role of SMAP data here in the algorithm? Any information from SMAP L3 is used?

Reference:

Response: Thank you very much for your comment. While as for SMAP L3 data, this Level-3 (L3) soil moisture product provides a composite of daily estimates of global land surface conditions retrieved by the Soil Moisture Active Passive (SMAP) passive microwave radiometer. SMAP L-band soil moisture data are resampled to a global, cylindrical 36 km Equal-Area Scalable Earth Grid, Version 2.0 (EASE-Grid 2.0). the website to download this data is :

https://nsidc.org/data/spl3smp/versions/8

There are two roles for the employed SMAP data, on one side, we can get the vegetation (VOD), surface roughness information so that we can get the pure Fresnel reflectivity for soil moisture retrieval, on the other side, we can compare the retrieval accuracy by comparing the FY-3E GNOS-R data and SMAP L3 data.

  1. From the retrieval point of view, it is ok to use the Fresnel reflection to account for the surface scatter. But remember that the physics behind SMAP observes and GNSS-R is different. Again, coherent scattering is one of the ways to interpret the data but it is less likely to be the real physics. You may refer to :
  2. M. Al-Khaldi, J. T. Johnson, S. Gleason, E. Loria, A. J. O’Brien,and Y. Yi, “An algorithm for detecting coherence in cyclone global navigation satellite system mission level-1 delay-Doppler maps,” IEEE Trans. Geosci. Remote Sens., early access, Aug. 11, 2020, doi: 10

And

  1. M. Al-Khaldi et al., "Inland Water Body Mapping Using CYGNSS Coherence Detection," in IEEE Transactions on Geoscience and Remote Sensing, vol. 59, no. 9, pp. 7385-7394, Sept. 2021, doi: 10.1109/TGRS.2020.3047075.

Response: Thank you very much, we have revised our paper according to your suggestions.

It should mention that most of the present works related to soil moisture retrieval using space-borne GNSS-R, such as CYGNSS have employed coherent scattering for the final retrieval, therefore, we also adopted this assumption when using FY-3E GNOS-R for soil moisture inversion. The following two references are added in the revision.

  1. M. Al-Khaldi, J. T. Johnson, S. Gleason, E. Loria, A. J. O’Brien,and Y. Yi, “An algorithm for detecting coherence in cyclone global navigation satellite system mission level-1 delay-Doppler maps,” IEEE Trans. Geosci. Remote Sens., early access, Aug. 11, 2020, doi: 10

And

  1. M. Al-Khaldi et al., "Inland Water Body Mapping Using CYGNSS Coherence Detection," in IEEE Transactions on Geoscience and Remote Sensing, vol. 59, no. 9, pp. 7385-7394, Sept. 2021, doi: 10.1109/TGRS.2020.3047075.

As for SMAP, the Fresnel reflection coefficients for horizontal and vertical polarization are::

                                               (1)

                                             (2)

While  is the incidence angle,  is the complex dielectric constant, the subscript h and v demonstrate the polarization state.

For completely smooth surfaces, the cross-polarization term  and can be ignored. For GNSS signals, the satellite transmission signal is RHCP, which can be understood as a linear combination of horizontal and vertical polarization components.

         (3)

Therefore, for the final retrieval, we will employ the above three equations to get the soil moisture results.

Round 2

Reviewer 1 Report

The authors significantly improved their manuscript. The manuscript could be accepted for publication. 

Author Response

Thank you very much for your comments。

Reviewer 2 Report

The manuscript has been improved a lot and most of the questions have been addressed. There is just 1 point I want to point out before the paper can be accepted. 

  1. Figure 5 (a) and (b) are not acceptable at all. Your receiver is not CYGNSS at all! You cannot just use other people’s figure without any modification. Please produce a figure by yourself or at least change the name of the receiver! 

I have no other comments. It will be appreciated if you can provide a reference discussing the signal processing flow.  

Bests

Author Response

Resposne:  Thank you very much for your comments and we have revised our paper according to your suggestions. While a new reference as you mentioned is added in the revision like the following:

黄芳,夏军,尹春,翟晓,杨,G.,白,W.,张,P.(2023)。FY-3E / GNOS-II上带有伽利略信号的星载GNSS反射计:测量,校准和风速检索。IEEE地球科学与遥感快报,20,1-5。https://doi.org/10.1109/lgrs.2023.3241358

同时,我们修改了图 5 中的错误,并提供了一个新图:

图5 .GNSS-R 散射几何 (a) 和镜面坐标系 (b)